# Crystal growth in confinement

Felix Kohler[1,2], Olivier Pierre-Louis [3] & Dag Kristian Dysthe [1]✉

The growth of crystals confined in porous or cellular materials is ubiquitous in Nature and forms the basis of many industrial processes. Confinement affects the formation of biominerals in living organisms, of minerals in the Earth's crust and of salt crystals damaging porous limestone monuments, and is also used to control the growth of artificial crystals. However, the mechanisms by which confinement alters crystal shapes and growth rates are still not elucidated. Based on novel in situ optical observations of (001) surfaces of $NaClO_3$ and $CaCO_3$ crystals at nanometric distances from a glass substrate, we demonstrate that new molecular layers can nucleate homogeneously and propagate without interruption even when in contact with other solids, raising the macroscopic crystal above them. Confined growth is governed by the peculiar dynamics of these molecular layers controlled by the two-dimensional transport of mass through the liquid film from the edges to the center of the contact, with distinctive features such as skewed dislocation spirals, kinetic localization of nucleation in the vicinity of the contact edge, and directed instabilities. Confined growth morphologies can be predicted from the values of three main dimensionless parameters.

Living organisms grow crystals to form bones, shells, coccoliths, and other complex biominerals. Confinement of these biogenic crystals during their formation allows for fine-tuned control of ionic composition, pH, and supersaturation[1] and permits to restrict of growth and to define crystal shapes[2,3]. Recent strategies for biomimetic materials design focus on the confinement of growing crystals by complex templates to control crystal morphology and strength[4,5]. In the Earth's crust, minerals also crystallize in confinement during diagenesis and metamorphism[6]. Furthermore, confinement is used to control the crystallization of ice[7], proteins[8], micro- and mesoporous crystals[9], and low-dimensional nanostructures[10]. Nevertheless, the understanding of the microscopic mechanisms by which nanoconfinement controls the morphology is still lacking.

As imaging methods reach a higher resolution, our view of crystal growth is evolving[7,11–14]. However, nanoconfined crystal growth remains largely unexplored because high-resolution measurement techniques such as scanning tunneling microscopy, atomic force microscopy[14] and traditional electron microscopy can only image open surfaces. The recent development of liquid cell electron microscopy has allowed the imaging of nucleation and growth of nanoparticles[11–13] and confined ice[7], but cannot image atomic step dynamics in confinement. Optical microscopy has been used with atomic-scale resolution for studies of unconfined crystal growth from solution in situ[15–17]. In nanoconfinement, reflection interference contrast microscopy (RICM) has previously reached a measurement precision of 2–30 nm[18].

We have used RICM with high-intensity LED illumination, a high-resolution camera, and image analysis to achieve for the first time sub-nanometer-resolution topography measurements of atomically flat $NaClO_3$ crystals growing in nanoconfinement. Our measurements allow us to analyze quantitatively the growth process and unravel a nanoconfinement regime that drives standard features of crystal growth into behaviors that are distinct from those of free surfaces[19,20]. Among these peculiar behaviors of nanoconfined growth, we observed skewed spirals, strong localization of nucleation along the contact edge, and instabilities such as fingering and bunching of molecular steps dictated by the orientation of and distance to the contact edge. Confined growth therefore proceeds with a specific growth mode characterized by the two-dimensional transport of growth units along the liquid film from the edge to the center of the contact, which produces gradients that control the growth of new molecular layers.

[1]The NJORD Centre, Department of Physics, University of Oslo, P.O. box 1048 Blindern, 0316 Oslo, Norway. [2]Expert Analytics, Møllergata 8, 0179 Oslo, Norway. [3]Institut Lumière Matière, Université de Lyon, Université Claude Bernard Lyon 1, CNRS, F-69622 Villeurbanne, France. ✉e-mail: dagkd@fys.uio.no

## Results and discussion

### Measurement of single molecular layers

We observe single $NaClO_3$ crystals growing from solution in a closed chamber with strictly controlled bulk solution supersaturation $\sigma = c/c_0 - 1$, where $c$ is the concentration and $c_0$ the equilibrium (saturation) concentration. The nanoconfined crystal growth is observed from the bottom and the distance $\zeta$ between the confining glass coverslip and the crystal surface is measured by reflection interference contrast microscopy (RICM) as shown in Fig. 1A–C. For distances $\zeta < 125$ nm the height relative to the local mean $z = \zeta - \zeta_0$ can be calculated from the image intensity with unprecedented, sub-nanometer precision. Figure 1D, E shows two crystal step fronts with a thickness of 3.3 Å relative to the time-averaged interface $\zeta_0 = 53$ nm. We found that on the (001) surface of $NaClO_3$ the minimum growth step height equals $z_0 = 0.33$ nm. This observation shows that $NaClO_3$ grows in an interlaced manner with two alternating monolayers, which differ in growth kinetics and which sum up to the thickness of one elementary cell height of $2z_0 = 0.66$ nm[21].

### Nucleation of molecular layers

In about 95% of our experiments, the crystals had no dislocations and we did not observe any growth for a supersaturation $\sigma < 0.048$. Above that threshold new layers nucleate on the confined facet and form two-dimensional monolayer islands that propagate until they cover the facet (see Fig. 2A and Supplementary Movies 1–3). On the length scale of our spatial resolution (300 nm) and time resolution (0.1 s) we observe that the growth steps at the edge of these monolayers flow unimpeded as if no solid was in direct contact with the growing crystal (see Supplementary Movies 1–5). A layer of spacer particles of diameter 10–80 nm is dispersed between the glass coverslip and the $NaClO_3$ crystal[22] (see Supplementary Fig. S2), both to mimic a rough contact and to control the distance $\bar{\zeta}$. Each time a new layer is added on the nanoconfined surface, the crystal surface is pushed back by the disjoining pressure and relaxes towards its equilibrium position (see Fig. 2B and Supplementary Fig. S3), raising the macroscopic crystal by

one molecular layer. A few molecular layers of fluid therefore appear to be sufficient for growth steps to propagate unaffected by the presence of spacer particles. The same effect of the disjoining pressure, but without spacer particles and with lower resolution, was observed for $CaCO_3$ crystals[23].

By counting the new layers (rapid drops of 0.66 nm in $\bar{\zeta}$ in Fig. 2B), we have measured the nucleation rate, $\tau_N^{-1}$, on the confined facet as function of supersaturation (see Fig. 2C). The standard theory of nucleation predicts that in the limit of small super-saturations $\sigma \ll 1$ corresponding to our experiments, the nucleation rate per unit area is $J = J_c(\sigma) e^{-\sigma_c/\sigma}$ (see the Supplementary Information). The critical supersaturation $\sigma_c$ can be expressed in terms of physical quantities $\sigma_c = \pi \Gamma^2 / 4z_0^2$, where $z_0$ is the crystal step height and $\Gamma$ is the molecular line tension length scale (see the Supplementary Information) and the critical nucleation rate $J_c$ depends only algebraically on $\sigma$. Fitting the nucleation rate expression to experimental data (see Fig. 2C) yields $\sigma_c = 1.1 \pm 0.1$ which leads to $\Gamma = 0.40 \pm 0.02$ nm.

We observe that at low nucleation rates, nucleation may occur anywhere on the confined surface (see Fig. 2D for $\sigma < 0.051$ and Supplementary Movie 1). As supersaturation and nucleation rate increase most molecular layers on the confined surface are nucleated close to the edge (see Fig. 2D for $\sigma > 0.051$). This abrupt change in nucleation localization is due to the depletion of ions in the confined fluid when a molecular layer grows. Diffusion does not have time to transport ions before the next nucleation event and a concentration gradient develops. Since nucleation depends exponentially on concentration, it localizes at the outer edge. The results in Fig. 2D agree with the nucleation theory we have developed for confinement (see Supplementary Information).

The number of monolayers of solid that can be formed by the ions in the fluid film is $\Theta_{eq} = \frac{\bar{\zeta}}{z_0} \frac{c_0}{c_s}$ ($= \bar{\zeta}/1.2$ nm for $NaClO_3$), where $c_0$ and $c_s$ are the molar densities of the fluid and solid. The coverage, $\Theta_{eq}\sigma$, the number of monolayers of solid that can be formed from the excess of ions in the supersaturated liquid is a relevant quantity for systems with

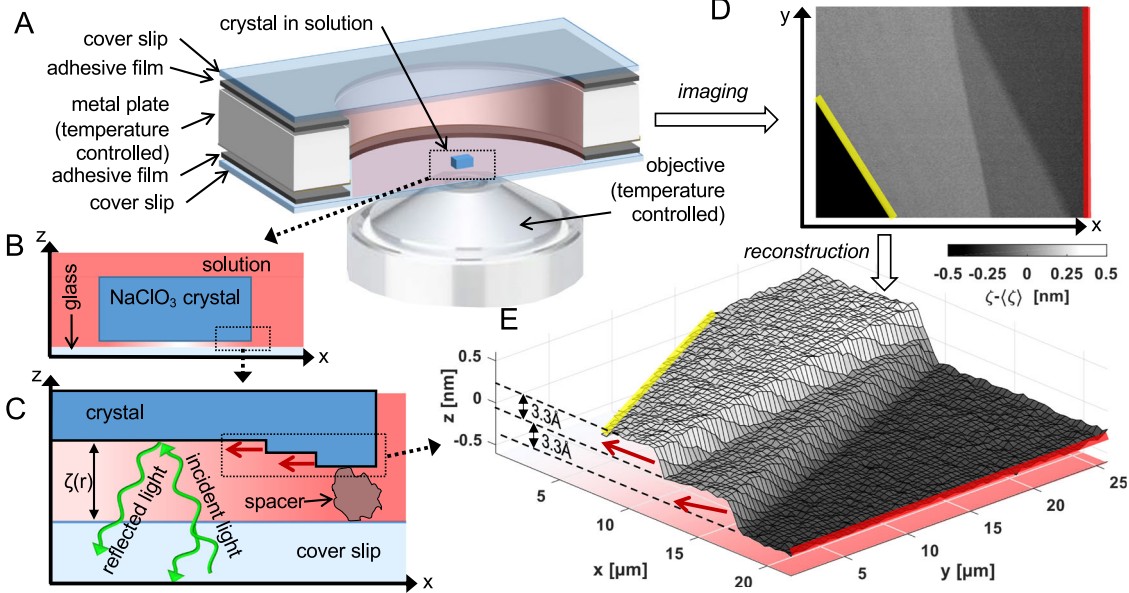

**Fig. 1 | Experimental setup. A** Sketch of experiment chamber with crystal in solution and high-resolution microscope objective. **B** Vertical cut of crystal. **C** Reflection of light from crystal and confining surface enables interferometric determination of the distance $\zeta(r)$ between the crystal and the glass surface and schematic of molecular steps of layered growth from outer edge. **D** Interferometric image of part of a growth rim of a crystal surface. **E** Reconstruction of local crystal height relative to the time-averaged interface $z = \zeta - \bar{\zeta}$, with $\bar{\zeta} = 53$ nm from the image intensity in (**D**). The measured 0.33 nm height of the steps corresponds to single molecular layers of the $NaClO_3$ crystal. **B, C, E** The color intensity indicates the solution supersaturation from high (red) in the bulk to zero (white) at the center. The red arrows indicate the growth direction of the steps as interpreted from the time-lapse movies S1–S5 in the Supplementary Information.

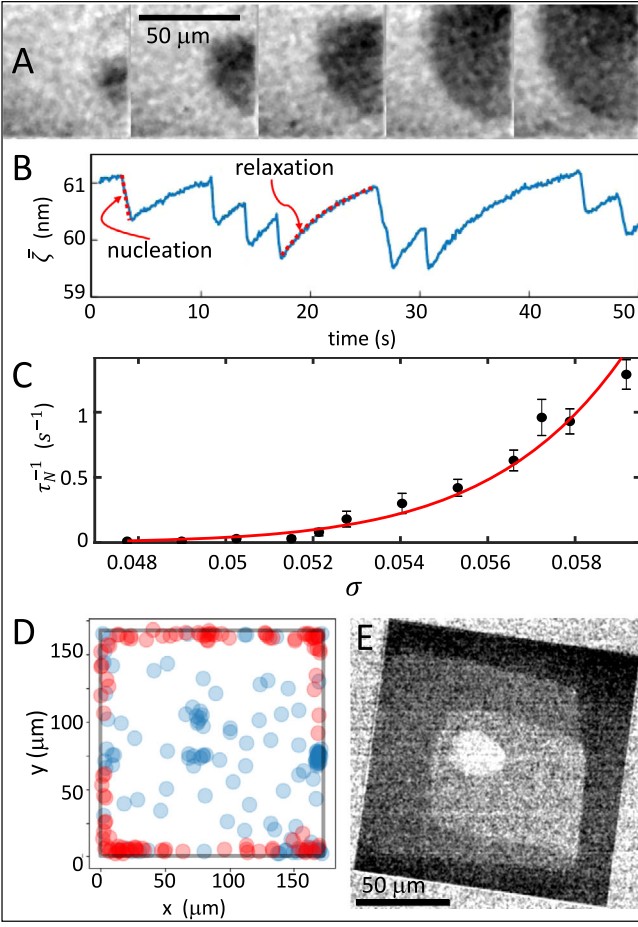

**Fig. 2 | Nucleation of molecular layers. A** Time series (0.1 s interval) of average subtracted RICM images of nucleation and spreading of a new layer (darker gray) at $\sigma = 0.051$. **B** Temporal evolution of mean distance, $\bar{\zeta}$, for a small crystal (L = 48 μm) showing sudden nucleation events (steep negative slopes of one unit cell height, 0.66 nm) followed by a relaxation towards equilibrium distance. **C** Nucleation rate as function of supersaturation. The nucleation rate was determined by counting of new layers in distance–time curves like (**B**). The error bars are standard deviations. The red line is a fit of nucleation theory to the data. **D** Localization of nucleation at different supersaturations. Position of nucleation events at $\sigma \leq 0.051$ (blue) showing random, homogeneous nucleation and a weak tendency of heterogeneous nucleation at three locations, and $\sigma > 0.051$ (red) showing strong localization at the edge. **E** Enhanced RICM image from Supplementary Movie 2 of crystal at $\bar{\zeta} = 53$ nm, $\sigma = 0.057$ showing four molecular layers (different gray levels) with step fronts of height 0.33 nm advancing inwards. Nucleation at the crystal edges and faster step front propagation in the vicinity of the crystal edge are caused by concentration gradients between the edges and the center.

two-dimensional mass transport such as surface diffusion in molecular beam epitaxy (MBE)[19]. Nanoconfined crystal growth is distinct from MBE in that the coverage can be larger than one. For the experiment in Fig. 2C, $\bar{\zeta} \approx 48$ nm and $\sigma = 0.05$, leading to $\Theta_{eq}\sigma \approx 2$. Due to the gradient in concentration, the layers spread fast along the edges and slower inwards towards the center of the confined surface as shown in Supplementary Movie 2. At higher supersaturations, $\sigma > 0.058$, multiple steps may nucleate before the layers have spread across the entire confined surface. Once a new layer is nucleated and spreading, other step fronts closer to the center slow further down because ions for growth are consumed by the outer step fronts and steps accumulate in a scenario similar to the bunching instability[20]. Then the step bunch stops, leaving a "cavity"[22] (light gray region in Fig. 2E, Supplementary Fig. S4, and Supplementary Movie 3) in the center of the crystal surface.

## Spiral growth and step flow velocity

In about 5% of the observed crystals, new molecular layers continuously originate from screw dislocations emerging at the surface like in Fig. 3 and Supplementary Movie 5 in the Supplementary Information, even at supersaturations far below the 2D nucleation threshold. The sharp jumps of intensity level are molecular step edges 0.33 nm in height each. The spiral atomic steps emerging from the dislocation are strongly skewed toward the edge of the contact: while steps moving toward the edge are accelerated, those flowing towards the inner part where a cavity is present are slowed down. One observes that there are two alternating types of step edges, one elongated in the vertical direction the other in the horizontal direction in the image, which correspond to the two half-unit cell interlaced layers.

In the yellow and blue sub-regions in Fig. 3C (a and b), we have measured the step flow propagation velocities of the two half layers along the outer boundaries where the solution supersaturation can be approximated by the supersaturation of the bulk solution. In the selected areas, the pixel values were averaged in a perpendicular direction to the border and filtered (Gaussian, standard deviation of kernel: s = 2) in the parallel direction. The step front was detected by finding the peak in the differentiated curve.

Measuring the step flow velocity as a function of orientation and position (see Fig. 3D and in the Supplementary Information) and assuming that supersaturation is proportional to the distance from the outer edge, the step velocity can be modeled as $v(\sigma, \theta) = \alpha \sigma k(\theta)$, where $k(\theta)$ is an anisotropy ratio. Simulations based on this law shown in Fig. 3F, G are in close correspondence with experiments in Fig. 3A–D when $\alpha = 1170 \pm 110$ μm/s with the anisotropy ratio reported in Fig. 3E.

The front velocities in Table 1 were measured as described in "Methods". Because there is a 180° rotation of the chlorate orientation between two half layers (A and B) of a unit cell[21] the velocity field of layer B correspond to the one of layer A rotated by 180°: $v_{A,-\vec{y}} = v_{B,\vec{y}}$ and $v_{A,\vec{y}} = v_{B,-\vec{y}}$. We define the kinetic anisotropy ratios $k_A(\theta) = v_A(\theta)/v_{B,\vec{x}}$ and $k_B(\theta) = v_B(\theta)/v_{A,\vec{y}}$, where the angle $\theta$ is relative to the positive $\vec{y}$-direction. In Fig. 3E, we show the kinetic anisotropy ratios, $k(\theta)$, of the fronts. Note that the vertical and horizontal velocities are in principle for straight steps with a vanishing topological kink site density. The step velocity is normally an increasing function of the kink site density as well resulting in a function $k(\theta)$ whose principal directions are indicated in Fig. 3E with a maximum velocity close to SE and NW, respectively.

There is a difference in supersaturation from the outer rim (red), $\sigma = 0.002$, to the edge of the cavity (light red), $\sigma \approx 0$. We have therefore measured the step front propagation velocity upwards in the image Fig. 3C in the green region c. Figure 3D shows the measured velocities of the step fronts, A and B, of the two half layers of the unit cell. The straight line fits demonstrate that the step front velocity is linear in vertical position, going to zero at the cavity and the maximum values (found in Table 1) at the outer edge.

In the very small range of supersaturations over the growth rim it is reasonable to assume that the supersaturation is linear in position, that is, the step velocity is linear in supersaturation:

$$v(\sigma, \theta) = \alpha \sigma k(\theta), \tag{1}$$

where $\alpha = 1170 \pm 110$ μm/s.

Using this relation, we have simulated the A and B step front propagation from a dislocation source situated at the inner rim edge like in Fig. 3A–C. The local step propagation velocity normal to the step front follows from (1). The supersaturation $\sigma$ is assumed to be a linear function of the distance from the outer rim edge and constant in time in this stationary state. Each point of the front is shifted with every time step according to the local concentration and the orientation-dependent growth kinetics as described by Eq. (1). This emulates a part of an extended growth rim, where the inner (lower) part is

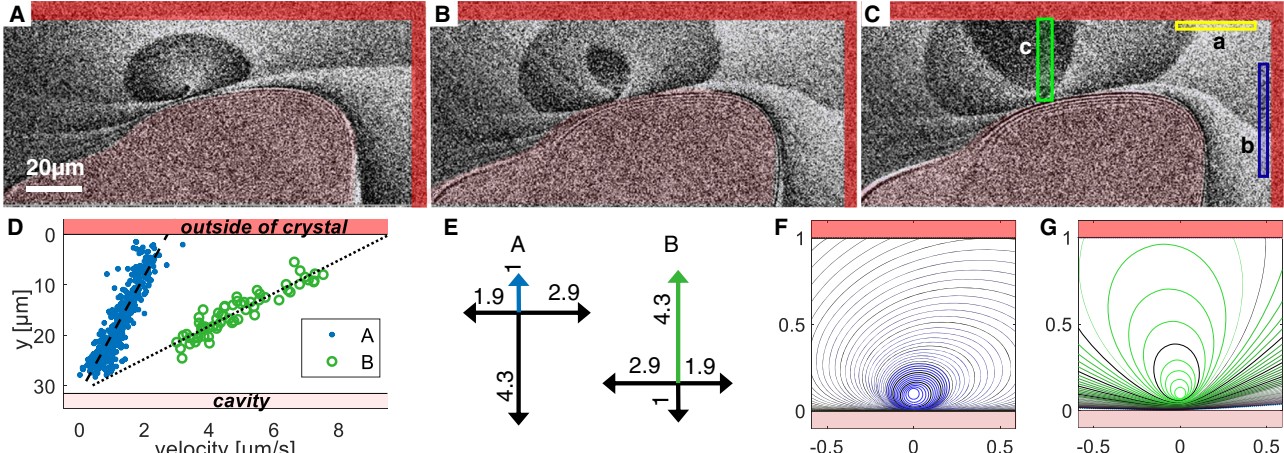

**Fig. 3 | Spiral growth in nanoconfinement. A–C** Average subtracted RICM image series at a bulk supersaturation $\sigma = 0.002$ with 3-s intervals. The areas outside the crystal and inside the cavity are marked in respectively red and light red colors. The oval regions of different intensities (gray levels) are molecular layers 0.33 nm in height each. The regions a and b in (**C**) have been selected for the determination of the step flow velocities along the outer rim boundaries. The green region c in (**C**) was selected to determine position (concentration) dependencies of step fronts shown in (**D**): position dependence of the step front propagation velocity in region c. The dashed and the dotted lines are linear fits to the data. **E** Kinetic anisotropy ratios of the two half layers A (blue) and B (green) corresponding to the fronts shown in (**D**). **F, G** Evolution of the A and B step fronts simulated by simple forward step algorithm using a linear supersaturation gradient and kinetic anisotropy ratios displayed in (**E**). One observes close correspondence with the step front shapes in the images (**A–C**).

connected to the cavity. For the orientation dependency of the step front kinetics, the experimental values are used. The A and B fronts have kinetic anisotropy ratios $k(\theta)$ as shown in Fig. 3E. The resulting A and B step fronts are shown in Fig. 3H, I for every 10th simulation time step. One observes the close correspondence with the step fronts in Fig. 3A–D. This shows that the simple assumptions of our simulations are valid for a crystal surface confined about 50 nm from an inert surface, with a small bulk supersaturation ($\sigma = 0.002$) and a growth spiral that ensures that step fronts pass with regular intervals (every $\approx 10$ s).

## Step flow instability

When the distance between the crystal and the substrate is small, we observe that the otherwise smooth step fronts can develop protruding "fingers" (see Fig. 4 and Supplementary Movie 4). The smooth, curved step first propagates along the facet edge and then inwards towards the facet center. At a distance $\approx 50\,\mu m$ from the edge, the step destabilizes with an initial wavelength $\Lambda \approx 12\,\mu m$. The tips of the fingers advance at a constant speed $55 \pm 5\,\mu m/s$ in the direction of maximum kinetic anisotropy, whereas the slowest parts of the step front slow down as $v(t) = v(t_0)\sqrt{t_0/t}$, where $v(t_0 = 1.8s) \approx 20 \pm 3\,\mu m/s$ is the velocity when the front destabilizes.

We interpret this instability to be a variant of the Mullins–Sekerka instability which leads to complex growth shapes like dendrites and snowflakes[19,24]: protuberances ahead of the step have a higher probability to catch randomly diffusing growth units (here ions), and thus grow faster than the other parts of the step. The wavelength $\Lambda = 2\pi\sqrt{3\ell\Gamma\Theta_{eq}}$ emerging from the instability accounts for a competition between this destabilizing point effect and the stabilizing effect of line tension, where $\ell = D/v_{step} \approx 20\,\mu m$ is the length scale of concentration gradients associated to the motion of the step at

velocity $v_{step}$. From $\bar{\zeta} = 20$ nm, we obtain a coverage $\Theta_{eq}\sigma = 0.9$ which agrees well with the fraction of the surface covered by the fingers. Using $\Gamma = 0.40$ nm as obtained from the nucleation rates, we find $\Lambda \approx 5\,\mu m$, which is of the same order of magnitude as the observed initial wavelength. Also, the velocity and radii of the fingertips are in agreement with existing theories for dendrite tips (see Supplementary Information). When $\bar{\zeta}$ is larger, the coverage is larger than one, and as suggested from model simulations in the literature[19,24] there is no instability as in Fig. 2.

## Confined growth morphologies

In order to reach a more general picture of confined growth modes and morphologies, we have also performed experiments on calcite crystals. The most important difference between $NaClO_3$ and $CaCO_3$ is that at room temperature the ratio of equilibrium solution concentration to solid concentration is $\frac{c_0}{c_s} \approx 0.3$ for $NaClO_3$ and $\frac{c_0}{c_s} \approx 2 \cdot 10^{-5}$ for $CaCO_3$. We have measured the step flow velocity on confined calcite surfaces (see Supplementary Fig. S6 and Supplementary Movie 6) and found that $\frac{v_s}{\sigma} \approx 10^{-8}$ m/s for $CaCO_3$, whereas $\frac{v_s}{\sigma} \approx 10^{-3}$ m/s for $NaClO_3$.

Even though solubilities and step flow velocities differ by orders of magnitude, we have observed[23,25] that calcite undergoes two morphological transitions that are the same or related to those we observe for $NaClO_3$:

- no cavity → cavity,
- stable growth → rough growth.

The transitions are related to the out of plane/in-plane growth rates outpacing diffusion that supplies growth units (molecules). Using the data from these two systems and the theoretical analysis of the growth morphologies, we can now summarize the processes, material constants and experimental parameters that govern the growth morphologies.

There are three processes that govern the observed phenomena: diffusion, with rate $D/L^2$, where $L$ is a characteristic length, step flow (in-plane growth), with velocity $v_s$ and nucleation with rate $JL^2$ (out of plane growth rate when step flow rate is fast). There are two material constants: solubility, $c_0/c_s$ and molecular/atomic step height, $z_0$. And there are three experimental parameters: radius (half length) of crystal, $L$, supersaturation, $\sigma$, and fluid film thickness, $\bar{\zeta}$. The latter parameter

**Table 1 | Step front velocities of the two interlaced layers A and B, measured at $\sigma = 0.002$ on the crystal shown in Fig. 3**

| | $v_A$, [µm/s] | $v_B$, [µm/s] |
|---|---|---|
| region a: $\vec{x}$-direction | 5.4 | 8.2 |
| region b: $\vec{y}$-direction | 2.8 | 12.2 |

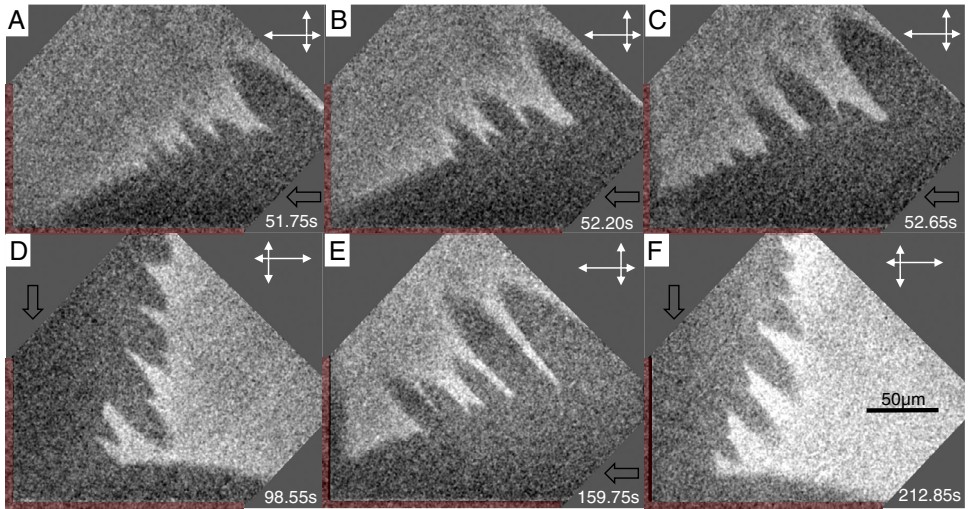

**Fig. 4 | Step front instability.** Average subtracted RICM images of corner of a $700 \times 700\ \mu m^2$ crystal with $\sigma = 0.053$ and $\bar{\zeta} = 22$ nm. Areas outside the crystal are marked in red. The dark areas correspond to a smaller distance to the confining glass and thus to the newly formed layer showing instabilities at the front.

**A–C** Time lapse of the same step front at 0.45 s intervals. **C–F** Four consecutive step fronts nucleated at opposite sides with a time delay of $54 \pm 7$ s. The black arrow indicates the direction from which the layer originates. The orientation-dependent growth kinetics of the respective layer is indicated by the white arrows.

we can combine with the material constants in the dimensionless parameter $\Theta_{eq} = \frac{\bar{\zeta}}{z_0} \frac{c_0}{c_s}$ (the number of solid monolayers that can be formed by ions in the fluid film).

We can reduce all these to three dimensionless numbers: nucleation rate over diffusion rate $JL^4/D$, step flow (convection) rate over diffusion rate, also called the Peclet number, $P = Lv_s/D$, and coverage, $\Theta_{eq}\sigma$, which is the confinement parameter.

The transition line from no cavity to cavity (measured both on NaClO$_3$[22] and CaCO$_3$ crystals[23,25]) was previously formulated as $u_z = 4\sigma\bar{\zeta}\frac{c_0 D}{c_s L^2}$[22] where $u_z$ is the out of plane (vertical) growth rate. When the step flow rate is not the limiting process $u_z = z_0 JL^2$. The condition for cavity formation can then be written as $4\Theta_{eq}\sigma < JL^4/D$.

The transition from a stable to an unstable front requires that the coverage is below 1 and that the step front outruns diffusion. In Supplementary Fig. S5 one observes that the instability at coverage $\Theta_{eq}\sigma = 0.9$ appears when the distance $L$ from the crystal edge is larger than the diffusion length $D/v_s \approx 50\ \mu m$. For calcite on the other hand, the coverage is always far below 1. But since the requirement for advancing the step is to grow one monolayer, the balance between diffusion and step velocity must be multiplied by the minimum concentration, that is the coverage. The transition line for stable/unstable step front is therefore $\Theta_{eq}\sigma = \frac{v_s L}{D} = P$, where $P$ is the Peclet number. We can now test this prediction on the calcite smooth to rough rim transition. Using the step flow velocity $v_s = 20 - 40$ nm/s at $\sigma = 0.6$, we have two experiments to compare with. In Figure 7 of ref. 25, one observes that a calcite crystal with $\zeta \approx 40$ nm goes through the transition as the crystal grows from $L = 4\ \mu m$ to $L = 8\ \mu m$, which corresponds to $\Theta_{eq}\sigma/P$ going from 1 to 0.5, thus crossing the blue line in Fig. 5. In ref. 26, the confinement is changed from $\bar{\zeta} = 600$ nm to $\bar{\zeta} = 40$ nm, triggering a rough growth rim. The change in $\bar{\zeta}$ changes $\Theta_{eq}\sigma/P$ from 3 to 0.2, once again moving the confined surface across the blue line in Fig. 5. It must be remarked that the nature of the instabilities are not the same for the fingering instability on NaClO$_3$ and the rim roughening of CaCO$_3$. The first one, studied in detail above is in a single-step regime, while on the calcite growth rim there are multiple steps. The latter transition is probably due to step bunching but detailed, high-resolution experiments will be necessary to clarify this. Both instabilities are triggered by competition between growth and diffusion in a confined environment and both transitions are where the ratio of rates are equal to the coverage which is the pertinent confinement parameter.

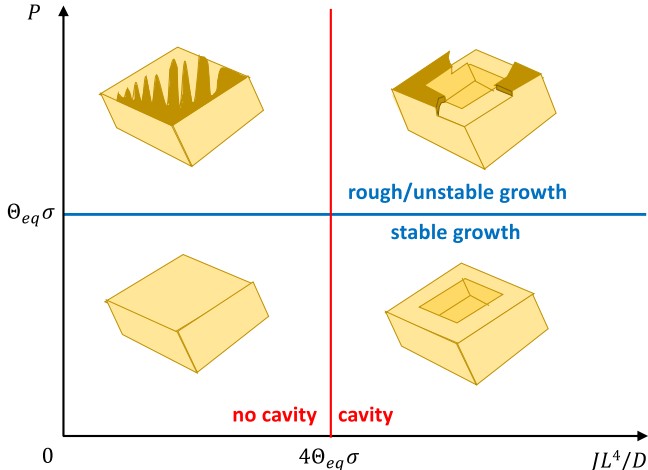

**Fig. 5 | Nonequilibrium morphology diagram of confined growth morphologies.** The stable step front morphologies (above the blue line) and the transition from no cavity to cavity (crossing the red line) have been observed both for NaClO$_3$ and CaCO$_3$ crystals. The finger-like instability to the upper left has only been observed on NaClO$_3$ and the growth rim roughening depicted to the upper right has been observed for both NaClO$_3$ and CaCO$_3$ crystals.

In order to calculate where in the growth morphology diagram (Fig. 5), a new system is placed, one can start by determining the experimental parameters $L$, $\sigma$, and $\bar{\zeta}$ (for the latter, use DLVO theory) and setting the nucleation rate, $J$ and step velocity $v_s$ to free surface values at the pertinent supersaturation. With the material constants at hand, the three dimensionless parameters $P$, $J/D$, and $\Theta_{eq}\sigma$ can be estimated.

## Conclusions

We have established that crystal growth steps can flow freely in confined interfaces unhampered by contact heterogeneities. The inherent transport restrictions in nanoconfinement cause a distinctive growth mode characterized by the two-dimensional transport of ions in the liquid film with gradients between the edge and the center of the contact that control the morphology and growth rate of nanoconfined

crystal surfaces. These gradients are constitutive and therefore unavoidable in nanoconfined growth and the free motion of molecular steps controlled by these gradients defines a nanoconfined growth mode that is different from known growth regimes. Some specific features associated to nanoconfined growth are the localization of nucleation along the contact edge, strongly skewed spirals, and control of the stability of molecular steps by the distance to the substrate.

Our observations demonstrate that high-resolution optical measurements of nanoconfined surfaces paves the way for in situ observation and identification of specific crystal growth regimes of relevance for biomineralization, templated crystal growth for advanced materials, and crystallization pressure.

The general growth morphology diagram, with dimensionless parameters that can be measured or estimated for any confined crystal growing from solution, can be applied to materials design, conservation, biomineralization, and geological crystal growth.

## Methods
We observe single $NaClO_3$ and $CaCO_3$ crystals growing from solution with strictly controlled bulk solution supersaturation $\sigma = c/c_0 - 1$, where $c$ is the concentration and $c_0$ the equilibrium (saturation) concentration.

### $NaClO_3$ experiments
As shown in Fig. 1A, the nanoconfined crystal growth is observed from the bottom using RICM. The glass coverslip shown on Fig. 1A–C supports the weight of the crystal and acts both as confining interface and as the reference mirror. The light reflected from the crystal and from the coverslip interfere and the intensity of the resulting image is a periodic function of the distance $\zeta$ between the two surfaces. For distances $\zeta < 125$ nm the height can be calculated from the image intensity with unprecedented, sub-nanometer precision (see the Supplementary Information) shown in Fig. 1D, E. This height profile reconstructed from the RICM image shows two-step fronts at the edge of growing atomic layers on the (001) facet with a thickness of 0.33 nm each.

### Sample preparation.
The sample is prepared under a laminar flow bench using pre-cleaned cover components. The chamber is filled with a defined (50 µl) volume of a saturated solution. A seed crystal of ~0.5 mm³ is added to the solution before the chamber is sealed. Before the actual experiment, the crystal is dissolved to a diameter of ~50 µm. This will dissolve all other small crystals in the chamber, which may accidentally appear and which can act as a seed of further crystallization processes. The high-nucleation barrier of $NaClO_3$ impedes the spontaneous nucleation of additional crystals during the experiment.

Because of the solubility of $NaClO_3$ is $10^4$ times higher than $CaCO_3$, the Debye screening length and crystal-glass distance $\bar{\zeta}$ at equal pressure is much smaller for $NaClO_3$ than for $CaCO_3$. In order to obtain a similar crystal-glass distance $\bar{\zeta}$ in the two sets of experiments, a layer of spacer particles of diameter 10–80 nm is dispersed between the glass coverslip and the $NaClO_3$ crystal[22] (see Supplementary Fig. S2).

### Temperature and imaging control.
The solubility, $c_0$ of $NaClO_3$ depends strongly on the temperature $T$ of the solution. Therefore, it is essential to carefully control the temperature of the sample. The lab was temperature controlled to $20 \pm 1°C$, the Olympus IX83 microscope was temperature controlled to $25 \pm 0.1°C$ by a Cube&Box temperature controller from Life Imaging Services (www.lis.ch), a temperature regulation fluid was controlled by a Julabo refrigerated−heating circulator to $25 \pm 0.01°C$ and flowed through a Peltier element heat exchanger on the oil immersion objective (Zeiss antiflex 63×/1.25) and to one side of Peltier elements on the crystal growth chamber. The other side of the Peltier elements was in direct contact with the metal of the crystal growth chamber. Thermistors were placed in the heat exchanger of the objective and in the metal of the chamber to enable temperature regulation and measurement. The heat flow between the temperature regulation fluid and the chamber was regulated by PID-controlled Peltier elements powered by in-house built current amplifiers. The PID, microscope, illumination, and camera controls were programmed in Matlab and Micromanager.

### Supersaturation of the solution.
The solubility $c_0(T)$ is the molar concentration $c$ of $NaClO_3$ in water in equilibrium with a $NaClO_3$ crystal. The solubility of aqueous $NaClO_3$ solutions depends strongly on the temperature $T$. In the temperature range, 0–50 °C, the dependence is linear and the relative change in solubility with respect to a reference temperature $T_0$ is $c_0(T)/c_0(T_0) - 1 = (T - T_0)/\delta T$, where $\delta T = 163$ K[27]. A crystal in the sealed chamber is allowed to equilibrate with the solution at temperature $T_0$. In order to determine $T_0$, the temperature is increased until the previously perfect cubic crystal starts to dissolve at its edges. Then the temperature is adjusted until neither growth nor dissolution at the roundish corners can be observed. Then the temperature is changed by $\Delta T = T_0 - T > 0$ to achieve the desired supersaturation $\sigma(T) = c/c_0(T) - 1 = c_0(T_0)/c_0(T) - 1 \approx \Delta T/\delta T + (\Delta T/\delta T)^2$. The temperature is controlled with accuracy $\pm 0.01$ K, thus $\sigma$ is controlled with accuracy $\pm 10^{-4}$. The growth of the crystal will consume ions from the solution and change the concentration (and supersaturation) of the solution. For the frequent case of crystal sizes $L$ smaller than $L = 200$ µm, the concentration of the solution in the chamber can be approximated to be constant. For larger crystals, i.e., crystals with edge lengths $L > 200$ µm, the concentration of the bulk solution is corrected by the consumption of material by the growing crystal.

### $CaCO_3$ experiments
The $CaCO_3$ crystal growth has been described in detail earlier[23,25]. The $CaCO_3$ crystal was nucleated and grown in a microfluidic channel with three inlets (I–III) meeting at the first junction and two more inlets (IV, V) meeting the main channel at the second junction. The channel dimensions are $120 \pm 2$ µm wide and 29 µm high, the length from the first to the second junction is $l_c = 50$ mm and the length from the second junction to the outlet is 10 mm. The channel networks were printed on a film substrate (Selba S.A, www.selba.ch). A photoresist (SU-8 GM1070, Gersteltec, www.gersteltec.ch) was spun on silicon wafers, UV radiated (UV-KUB2, http://www.kloe.fr) and developed with PGMEA (www.sigmaaldrich.com) according to producers data sheet. Channel networks were cast in PDMS (Dow Corning Sylgard-184A, www.sigmaaldrich.com) with a 1:10 elastomer to curing-agent ratio. Inlets and outlets of 1.5 mm diameter were punched subsequently and both the PDMS and glass covers were treated with corona plasma (Electro-Technic Model BD-20V, http://www.electrotechnicproducts.com) before assembly. The fluid flow into each channel was controlled by a gas pressure control system that includes an Elveflow controller (Elveflow OB1 mk3, www.elveflow.com) with flow rate control mode, flow valves (Elveflow MUX) and flow sensors (0.4–7 µL/min, Elveflow).

The fluids were injected in the five inlets (I−V) with the following fluid concentrations I: 2 mM $Na_2CO_3$, II: water, III: 2 mM $CaCl_2$, IV: 10 mM $Na_2CO_3$, V: 10 mM $CaCl_2$. The $CaCl_2$, $H_2O$, and $Na_2CO_3$ solutions mix in the main channel by diffusion and the relative flow rates determine the final $CaCO_3$ concentration. To induce nucleation, we use inlets IV and V to produce nuclei that attach to the walls of the channel. Inlets IV and V are used for nucleation only and are closed during growth. Multiple nucleations or nuclei at undesired locations are dissolved by lowering the concentration of the solution. Calcite nucleation and dissolution of nuclei is repeated until a nucleus is attached on the PDMS membrane in the desired region. After nucleation, a $CaCO_3$ concentration of $0.801 \pm 0.002$ mM has been used, which corresponds to a supersaturation of $\sigma = 0.6$[25].

In the $CaCO_3$ experiments, there are no spacers (as shown in Fig. 1C) because the crystals are so small that they are kept at a mean distance $\bar{\zeta} = 20 - 50$ nm by the double layer repulsion[23,25].

The $CaCO_3$ experiments were performed on an Olympus GX71 microscope with a green LED light source with a wavelength of 550 nm (from ThorLabs www.thorlabs.com) and UPLanFI 40×/0.75p objectives (from Olympus www.olympus-lifescience.com). Images were recorded using a Pointgrey camera (Mono, Grasshopper3, GS3-U3-91S6M-C, www.ptgrey.com) with 3376 × 2704 resolution.

### Reflection interference contrast microscopy

The crystals are observed using reflection interference contrast microscopy (RICM), which is based on the interference of the light reflected by the sample with the light reflected by the glass surface the sample is placed on[28]. This technique is mainly used to examine the contacts between biological cells and glass surfaces. Several improvements to this technique have been made including numerical analysis, which considers a finite illumination aperture and a finite Numerical aperture[18,29–31] the usage of the so-called antiflex technique to improve the signal-to-noise ratio[32], dual-wavelength reflection interference contrast microscopy[33] and the usage of a thin coating to shift the contact area away from the first minimum in intensity[34]. RICM has mainly been developed and used for measurements of absolute fluid film thickness of soft and biological matter and has reached a measurement precision of 2–30 nm[18,31,33,35,36].

A sketch of the principle of crystal growth measurements using RICM is shown in Fig. 1. The intensity

$$I = 2\pi \int_0^{\theta_{max}} \sin(\theta) I_0(\theta) \gamma(\theta) r(\theta) d\theta \qquad (2)$$

detected at a pixel of the detector results from the angular spectrum $I_0(\theta)$ of the illumination, the optical response function of the system $\gamma(\theta)$ and an interference-based reflectance factor $r(\theta)$. Since we use an objective with a high-numerical aperture, a large part of the light enters with a non-negligible angle $\theta$ in respect to the optical axis. Rotational symmetry is used in Eq. (2). The reflectance factor

$$r(\theta) = R_{g,s}(\theta) + R_{s,c}(\theta') + 2\sqrt{R_{g,s}(\theta)R_{s,c}(\theta')}$$
$$\cos\left(4\pi n_s \phi(\theta,\theta') \frac{\zeta(r)}{\lambda} + \pi\right) \qquad (3)$$

is a function of the local distance $\zeta(r)$ between crystal and glass interface, the wavelength $\lambda$, the refractive indices of the glass ($n_g = 1.52$), the crystal ($n_c = 1.515$) and the $NaClO_3$ solution ($n_s(c) = 1.32415 + 0.136 \frac{c}{100g+c}$, where $c$ is the concentration in g per 100 g $H_2O$[37]). $\phi(\theta,\theta') = \frac{1}{\cos(\theta')} - \frac{n_s}{n_g}\tan(\theta')\sin(\theta) \leq 1$ denotes a relative phase difference, which depends on the angles $\theta$ between the optical axis and the beam in glass and $\theta' = \arcsin\left(\frac{n_g}{n_s}\sin(\theta)\right)$ between the optical axis and the beam in solution. The reflectances $R_{g,s}(\theta)$ of the glass-solution interface and $R_{s,c}(\theta')$ of the solution-crystal interface are determined by the Fresnel equations using the respective refractive indices[37,38]. We expect that the resulting intensity–distance relation in the region from contact to the first maximum is well represented by assuming a uniform illumination ($I_0(\theta) = const.$) and system response ($\gamma(\theta) = const.$) in combination with an effective numerical aperture or maximal angle. In order to obtain this effective numerical aperture, we imaged the interference intensity from a spherical lens (Thorlabs, LA1540, focal length: f = 15 mm, radius 7.7 mm) in contact with the coverslip in a saturated $NaClO_3$ solution. The intensity–distance relation for the thus obtained calibration measurement is shown in Supplementary Fig. S1. We selected the effective numerical aperture to match the position of the first maximum determined by the calibration

measurement, i.e., NA = 1.23. Since the angular spectrum $I_0(\theta)$ of the measurement system does not depend on the reflectivity and thus the material, this value is also valid for the crystal experiments. In order to reconstruct the distances $\zeta$ between the crystal and the coverslip below the first maximum of the interference, we use the min/max method by Limozin et al.[31] in combination with the calibration described above. In the $NaClO_3$ experiments, a ThorLabs SOLIS-525C LED with a centroid wavelength of 525 nm has been used in combination with a 16-bit Andor Zyla 5.5 sCMOS camera (http://andor.oxinst.com). The glass coverslips used have an RMS roughness of 0.2 nm.

## Data availability

The image data generated in this study have been deposited in the Figshare database under the accession code https://doi.org/10.6084/m9.figshare.21201737. The laboratory notebook data are available under restricted access for privacy reasons, access can be obtained by the corresponding author on reasonable request.

## Code availability

Computer codes used to analyze data during the current study are available from the corresponding author on reasonable request.

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

## Acknowledgements

This project has received funding from the European Union's Horizon 2020 research and innovation program under the Marie Sklodowska-Curie Grant Agreement No. 642976 (ITN NanoHeal) and from the Norwegian Research Council Grant No. 222386. We thank Henrik Sveinsson for preparing MD configurations for Supplementary Fig. S2.

## Author contributions

F.K. designed and performed experiments, analyzed data, and wrote the manuscript. O.P.L. developed theory and wrote the manuscript. D.K.D. designed experiments, analyzed the data, and wrote the manuscript.

## Competing interests

The authors declare no competing interests.
