## [Peer Review File · Nature Communications]

Crystal growth in confinementREVIEWER COMMENTS

Reviewer #1 (Remarks to the Author):

The paper by Kohler et al. is well conceived, executed and written. It tackles the fundamental distinctions between crystal growth in a confined volume and crystal growth in a free environment. This issue relates to a broad range of areas: biomineralization, geology, nanotechnology, construction, archeology, and many others. The main challenge has been to observe crystal surfaces at molecular resolution in the vicinity of a wall, which interferes with the common scanning probe and optical techniques. Kohler et al. employ an ingenious method, reflection interference contrast microscopy, which allows them to observe crystal faces positioned a few nanometers from a glass wall and to accurately measure the characteristics of the gap (width and tilt) between the crystal surface and the glass. They detect steps as low as 3.3 Å on the crystal surface and monitor their growth and dissolution with high time resolution. Importantly, the used optical method is non-intrusive. They impose supersaturation by lowering the temperature and use the known temperature dependence of the solubilities of their two model crystal forms, sodium chlorate and calcite, to accurately evaluate the supersaturation. The authors accurately measure rates of 2D nucleation of new layers and record the locations of the nucleation events, they measure step velocities and correlate them with the non-uniform supersaturation along the crystal face in the vicinity of the wall, they observe two instabilities: of the step front, analogous to the classical 3D Mullins-Sekerka instability of crystal faces growing in a temperature gradient, and cessation of growth at the poorly supplied facet centers, leading to hollow growth. Another appealing feature of the paper is the combination of results for two systems that dramatically differ in solubility and crystal growth anisotropy. They provide a stability diagram that summarizes the combination of parameters that may lead to stable faced growth, hollow growth, step fingers, or combination of the latter two. A particularly appealing feature of the paper is the integration with several levels of theory: stability analyses, solute transport simulations, nucleation and crystal growth kinetic analyses.

This study represents a spectacular achievement of experiment and theory in materials synthesis and certainly deserves to be published in Nat. Comm. I have a few comments that only address the presentation and not the essence of the results.

My general comment is that the papers to have been written for readers who are experts in RICM and are familiar with the authors' previous papers.

p. 1 supersaturation σ is not defined. It is claimed that it is precisely controlled (maybe use strictly instead) but no information is provided which supersaturation the authors have in mind: in the solution bulk away from the crystal, close to the crystal edge, in the center of the face?

Imposing supersaturation by lowering the temperature offers many advantages. Varying temperatures, however, may dramatically change the kinetic parameters and introduce unnecessary complexity in the results interpretation. Please specify why you assume that the temperature ranges of the experiments are sufficiently narrow to minimize this effect.

The diagram in fig. 5 is one of the best features of the paper; in its current form, however, it is very hard to understand. Please clarify its parameters. What is the physical meaning of the ratio τ_D/τ_N ? Of the product $\theta_{eq} \sigma$? Why a diagonal line in these coordinates separates regions of stability from those of unstable growth? What is P and how does one choose its critical value? Why $P/4$ is important?

Fig. 3 is introduced before Fig. 2. I suggest splitting Fig 3 into two parts and placing the first of the new figures before Fig. 2, or as new panels in Fig. 2.

Classical, not standard nucleation theory.

The crystal surface relief is deduced from the interference between light reflected from it and

light reflected by the cover glass. Please specify the requirements for the flatness of the cover glass needed to obtain depth resolution as high as 3 Å.

Reviewer #2 (Remarks to the Author):

These are beautiful experimental results using reflection interference contrast microscopy to image the growth of confined NaClO₃. A large part of the paper is focused on describing the experimental set-up, which is justified given the unprecedented resolution of the experiments. My impression of this paper is that they have nice data, but they are struggling in the area of defining truly unique features associated with confinement. I mean, they do list features associated with crystal growth in confinement. However, I think such features would also arise without confinement and I feel challenged to understand why these features arise uniquely from confinement. Can the authors be more plain about what new features arise in confinement and why? I would also like to see some discussion on how different types of confinement might allow one to control these features to produce unique crystal morphologies.

Reviewer #3 (Remarks to the Author):

Title: Crystal growth in confinement

Authors: Kohler et al

The authors have for the first time examined the effect of nanoconfinement on crystal growth using

Reflection Interference Contrast Microscopy at sub-nanometer resolution. Their work on NaClO₃ salt has led to the discovery of several new phenomena particularly dictated by the two-dimensional transport of materials. Some of these unique features are localization of crystal nucleation along contact edges, fingering instabilities, formation of skewed spirals and so on. The experimental details, observed data and analysis of observations have been presented elaborately. Looking at the significance of the results presented, this manuscript is recommended for publication in its current form.

Answers to reviewer comments

Reviewer #1 (Remarks to the Author):

The paper by Kohler et al. is well conceived, executed and written. It tackles the fundamental distinctions between crystal growth in a confined volume and crystal growth in a free environment. This issue relates to a broad range of areas: biomineralization, geology, nanotechnology, construction, archeology, and many others. The main challenge has been to observe crystal surfaces at molecular resolution in the vicinity of a wall, which interferes with the common scanning probe and optical techniques. Kohler et al. employ an ingenious method, reflection interference contrast microscopy, which allows them to observe crystal faces positioned a few nanometers from a glass wall and to accurately measure the characteristics of the gap (width and tilt) between the crystal surface and the glass. They detect steps as low as 3.3 Å on the crystal surface and monitor their growth and dissolution with high time resolution. Importantly, the used optical method is non-intrusive. They impose supersaturation by lowering the temperature and use the known temperature dependence of the solubilities of their two model crystal forms, sodium chlorate and calcite, to accurately evaluate the supersaturation. The authors accurately measure rates of 2D nucleation of new layers and record the locations of the nucleation events, they measure step velocities and correlate them with the non-uniform supersaturation along the crystal face in the vicinity of the wall, they observe two instabilities: of the step front, analogous to the classical 3D Mullins-Sekerka instability of crystal faces growing in a temperature gradient, and cessation of growth at the poorly supplied facet centers, leading to hollow growth. Another appealing feature of the paper is the combination of results for two systems that dramatically differ in solubility and crystal growth anisotropy. They provide a stability diagram that summarizes the combination of parameters that may lead to stable faced growth, hollow growth, step fingers, or combination of the latter two. A particularly appealing feature of the paper is the integration with several levels of theory: stability analyses, solute transport simulations, nucleation and crystal growth kinetic analyses.

This study represents a spectacular achievement of experiment and theory in materials synthesis and certainly deserves to be published in Nat. Comm. I have a few comments that only address the presentation and not the essence of the results.

My general comment is that the pers papers to have been written for readers who are experts in RICM and are familiar with the authors' previous papers.

1.1 p. 1 supersaturation σ is not defined. It is claimed that it is precisely controlled (maybe use strictly instead) but no information is provided which supersaturation the authors have in mind: in the solution bulk away from the crystal, close to the crystal edge, in the centre of the face?

We have changed "supersaturation σ precisely controlled" to "strictly controlled bulk solution supersaturation $\sigma = c/c_0 - 1$, where c is the concentration and c_0 the equilibrium (saturation) concentration". Further details of the supersaturation control is given in the third subsection of "Methods".

1.2 Imposing supersaturation by lowering the temperature offers many advantages. Varying temperatures, however, may dramatically change the kinetic parameters and introduce unnecessary complexity in the results interpretation. Please specify why you assume that the temperature ranges of the experiments are sufficiently narrow to minimize this effect.

There is a long tradition for using temperature to control supersaturation in crystal growth studies for NaClO₃ and many other crystals. For example, temperature differences of up to 5 K were used to control supersaturation in a study of the growth kinetics of NaClO₃ without

showing any visible effect on the growth rate (R. Ristic, J. N. Sherwood, and K. Wojciechowski, J. Phys. Chem 1993, 97, 10774). The largest supersaturation in our study (the experiments showing step flow instability) corresponds to a temperature change no larger than 10 K. Such a temperature change changes the self diffusion coefficient of water by about 25% and we expect the step flow velocity to change by a similar magnitude. The cavity formation has been shown for calcite where we do not use temperature for supersaturation control. The step flow instability depends on the ratio v_s/D which more or less cancels the temperature effects on v_s and D . We therefore have no reason to assume that there are dramatic changes in the kinetic parameters that affect the nature of our results. We have added the above text to the Supporting Information.

1.3 The diagram in fig. 5 is one of the best features of the paper; in its current form, however, it is very hard to understand. Please clarify its parameters.

We thank the reviewer for pushing us to improve this figure. In the process of reformulating the text we found an even simpler way to depict the morphology transitions.

1.4 What is the physical meaning of the ratio τ_D/τ_N ? Of the product $\theta \sigma$?

We have introduced the basic and derived quantities in a more logical fashion and explained better the significance of the dimensionless numbers.

$\tau_D/\tau_N = JL^4/D = \text{nucleation rate}/\text{diffusion rate}$. When this ratio increases beyond a limit ($4\theta \sigma$), the diffusion of growth units is not sufficient to feed the growth in the entire crystal plane and a cavity is formed. $\theta \sigma$ is the number of crystal layers that can be formed by the excess (beyond saturation) number of growth units in the confined fluid film, thus how much the crystal can grow without resupply by diffusion from the outside.

1.5 Why a diagonal line in these coordinates separates regions of stability from those of unstable growth?

We have now made a more rational diagram with rate ratios on both axis. The diagonal separation line in the former diagram is now represented by the peculiar result that the coverage determines both transitions. This is due to the fact that this is the only remaining dimensionless parameter in the system and it is the parameter that characterizes the confinement.

1.6 What is P and how does one choose its critical value?

We have added a short explanation of how to estimate all the dimensionless numbers for a new system.

1.7 Why $P/4$ is important?

The factor 4 (now on the horizontal axis of the diagram) is a geometrical factor that arises from the cylindrical geometry used to derive the cavity criterium. For a long and thin crystal this factor would be 1. For further details, see Kohler et al, PRL 121, (2018).

1.8 Fig. 3 is introduced before Fig. 2. I suggest splitting Fig 3 into two parts and placing the first of the new figures before Fig. 2, or as new panels in Fig. 2.

The reviewer's comment pointed out to us that we had repeated the explanation about measuring step front velocities twice and that the description was not general, but specific for the section "Spiral growth and step flow velocity". We have therefore moved the necessary description from "Methods" to "Spiral growth and step flow velocity". Then the figures are referred to in the correct order.

1.9 Classical, not standard nucleation theory.

This has been corrected.

1.10 The crystal surface relief is deduced from the interference between light reflected from it and light reflected by the cover glass. Please specify the requirements for the flatness of the cover glass needed to obtain depth resolution as high as 3 Å.

The results presented in this paper are mainly based on

A The distance and inclination of the crystal relative to the glass interface

B The detailed surface relief of the crystal, e.g. step fronts appearing on the confined crystal interface

For most cases, it was possible to observe the crystal at a very low supersaturation with perfect interfaces. At this conditions, we are assuming that we observe flat crystal interfaces. Thus, the distance and position (A) can be determined using a larger area, which leads to an averaging over roughness. The detailed surface relief (B) can be determined as deviation from the perfect crystal interface and thus eliminating the effects of cover slip roughness, which has been measured to be 0.2 nm.

We have added "The glass coverslips used have an RMS roughness of 0.2 nm." in the text.

Reviewer #2 (Remarks to the Author):

These are beautiful experimental results using reflection interference contrast microscopy to image the growth of confined NaClO₃. A large part of the paper is focused on describing the experimental set-up, which is justified give the unprecedented resolution of the experiments. My impression of this paper is that they have nice data, but they are struggling in the area of defining truly unique features associated with confinement. I mean, they do list features associated with crystal growth in confinement. However, I think such features would also arise without confinement and I feel challenged to understand why these features arise uniquely from confinement.

2.1 Can the authors be more plain about what new features arise in confinement and why?

As stated in the report of Reviewer #3, we observe growth regimes that are limited by two-dimensional (2D) diffusion between in the liquid film between the crystal and the substrate. This 2D character of the transport process is exacerbating phenomena that are governed by diffusion-limited mass transport. Indeed, in classical 3D diffusion limited dynamics, specific features such as the Mullins-Sekerka instability occur because growth units diffusing from the bulk of the solution have a higher probability to reach first the protruding geometric features of the crystal, such as corners and facet edges. However, in 3D, growth units can still diffuse around corners and facet edges to reach the center of the facet. This is not true anymore in our 2D confinement regime. Indeed, when diffusing along the liquid film in the contact zone, growth units start from the facet edges and have to pass in the vicinity of atomic steps which are sinks during growth before reaching he center of the facet. Hence, due to their 1D character they also separate the contact region into distinct 2D regions where they impose boundary conditions (This is to some extent similar to the crucial role of steps in Molecular Beam Epitaxy (MBE) in the presence of surface diffusion. However, growth units are deposited everywhere on the surface in MBE, while they only come from the edges of the contact zone in confined growth.). In addition, the film contains much less growth units than a bulk liquid (this effect is quantified by the dimensionless parameter $\Theta_{eq} \sigma$) so that growth leads to a huge depletion effect on this concentration field.

As a consequence of this peculiar growth mode, with strong influence of atomic steps and very large depletion effects, the spatial variations of the growth unit concentration fields in this 2D mass transport regime are strongly enhanced, leading to the spectacular effects that we observe: skewed spirals, cavity formation, localization of nucleation, etc.

2.2 I would also like to see some discussion on how different types of confinement might allow one to control these features to produce unique crystal morphologies.

This is the question addressed in Fig. 5 and in the associated discussion in the manuscript. We have improved the figure and rephrased the discussion to clarify the key physical mechanisms at play and their consequences on confined growth.

Reviewer #3 (Remarks to the Author):

Title: Crystal growth in confinement

Authors: Kohler et al

The authors have for the first time examined the effect of nanoconfinement on crystal growth using Reflection Interference Contrast Microscopy at sub-nanometer resolution. Their work on NaClO₃ salt has led to the discovery of several new phenomena particularly dictated by the two-dimensional transport of materials. Some of these unique features are localization of crystal nucleation along contact edges, fingering instabilities, formation of skewed spirals and so on. The experimental details, observed data and analysis of observations have been presented elaborately. Looking at the significance of the results presented, this manuscript is recommended for publication in its current form.

Some additional changes to the manuscript:

We have changed the use of "slow and fast fronts" to "step fronts, A and B, of the two half layers of the unit cell" and "A and B fronts"

This is because both fronts attain the same velocities, but at different orientations. They are only "fast" and "slow" in any give direction.

REVIEWERS' COMMENTS

Reviewer #1 (Remarks to the Author):

The authors have properly addressed all comments by myself and the other reviewers. The manuscript can now be published in Nature Communications

Reviewer #2 (Remarks to the Author):

The authors have adequately addressed my concerns.

Reviewer #3 (Remarks to the Author):

The manuscript is acceptable in its revised form. There is no further comment to address.